# STRUCTURED RETRIEVAL-AUGMENTED GENERATION FOR MULTI-DOC MULTI-ENTITY QUESTION ANSWERING

## ABSTRACT

Multi-document Multi-entity Question Answering (MDMEQA) fundamentally requires models to track and connect the implicit logic between multiple entities across documents, a task that reveals critical limitations of Large Language Models (LLMs) and Retrieval-Augmented Generation (RAG) frameworks: they struggle to construct effective cross-document evidence chains and deduce entity relationships when faced with fragmented information. Although RAG improves answering capabilities through context injection, its coarse-grained retrieval strategy that relies on vector similarity often leads to the omission of critical facts. Meanwhile, graph-based RAG fails to efficiently integrate scattered complex relationship networks in multi-document scenarios, resulting in low efficiency in retrieving and reasoning MDMEQA. We propose **Structured Retrieval-Augmented Generation** (SRAG): a two-stage framework that first transforms unstructured text into semantically coherent relational tables via a SQL-driven Extraction-Retrieval module, then guides LLMs toward schema-aware relational reasoning over structured representations. This architectural breakthrough offers three key advantages: (1) SQL-powered indexing enables precise fact localization; (2) relational tables naturally support multi-hop entity join operations; (3) the structuring process mitigates the attention diffusion effect of LLMs. To verify the effectiveness of our proposed method, we evaluate SRAG on two multi-document QA benchmarks, MEBench and Loong. The results show that SRAG significantly outperforms the current state-of-the-art long-context LLMs and RAG systems, achieving 27.2% and 27% improvements in accuracy respectively. These results highlight the importance of structured data representation in enhancing complex reasoning and answer precision in multi-document multi-entity question answering. The source code and data have been made available at https://anonymous.4open.science/r/SRAG-07A7.

## 1 INTRODUCTION

Recent progress in Retrieval-Augmented Generation (RAG) has enhanced how language models access external knowledge, improving applications like question answering (Fan et al., 2024). Most QA systems focused on single-document scenarios, where the scope of information and entity relationships is constrained. However, real-world decision-making, such as synthesizing findings across multiple clinical studies, analyzing cross-company financial reports, or summarizing legal precedents spanning dozens of case files, demands a more advanced paradigm: Multi-document Multi-entity QA (MDMEQA), which requires models not only to retrieve information from disjoint document collections but also to track and connect implicit logical relationships between multiple entities. This cross-document, multi-entity reasoning requirement exposes critical gaps in state-of-the-art technologies, including LLMs and RAG frameworks.

LLMs—even long-context variants—struggle with MDMEQA due to two inherent limitations. First, their reliance on parametric memory leads to "attention diffusion": when processing fragmented information across

documents, the model's attention mechanism fails to prioritize and connect relevant entity relationships, resulting in incomplete or erroneous reasoning. Second, LLMs lack explicit mechanisms to construct cross-document evidence chains; they often treat each document in isolation, missing dependencies between entities that span multiple documents. RAG frameworks, which augment LLMs with external retrieval to address parametric memory limits, partially alleviate these issues but introduce new bottlenecks. Traditional RAG relies on coarse-grained vector similarity for retrieval: by embedding queries and document chunks into dense vector spaces, it prioritizes semantic similarity over real entity relationships, frequently omitting critical facts tied to specific entities. Graph-based RAG variants, which model entity relationships as graphs, fare no better in MDMEQA: multi-document scenarios generate scattered, overlapping relationship networks, making graph traversal inefficient and reasoning over multi-hop entity paths computationally prohibitive.

To address these limitations, we propose SRAG, a two-stage framework designed explicitly for MDMEQA. SRAG 's core insight is that structured data representation , rather than unstructured text chunks or sparse graphs, can bridge the gap between retrieval precision and multi-entity reasoning. The framework operates in two phases: (1) A SQL-driven Extraction-Retrieval module converts unstructured text from multiple documents into semantically coherent relational tables, where rows represent entity instances and columns encode attributes or relationships (*e.g.,* a table for "ResearchPapers" with columns "Author", "Institution", and "Cited"); (2) A schema-aware LLM reasoning module leverages the structure of these tables to perform targeted reasoning, using SQL as a "reasoning scaffold" to join tables across entities and mitigate attention diffusion. This design delivers three key advantages over existing methods: (1) SQL-powered indexing enables precise fact localization, as queries can directly target entity attributes rather than relying on imprecise vector similarity; (2) Relational tables natively support multi-hop entity joins, simplifying cross-document relationship inference; (3) The structuring process constrains the LLM's attention to relevant table columns and rows, eliminating noise from unstructured text and reducing attention diffusion.

To validate SRAG 's effectiveness, we evaluate it on two recent multi-document benchmarks: MEBench (Lin et al., 2025) and Loong (Wang et al., 2024). Results show that SRAG outperforms state-of-the-art long-context LLMs and RAG systems by significant margins: it achieves 27.2% higher accuracy on MEBench and 27% higher accuracy on Loong. These improvements confirm that structured data representation is a critical enabler of complex reasoning in MDMEQA, addressing the retrieval imprecision and reasoning inefficiency of prior approaches. These findings underscore the critical importance of structured data representation for complex, multi-faceted reasoning.

**Contributions**    Our notable contributions are summarized as follows.

- **Proposing the SRAG framework for MDMEQA**: We introduce a novel two-stage SRAG framework specifically designed to address the challenges of MDMEQA. SRAG integrates a SQL-driven Extraction-Retrieval module to convert unstructured text into relational tables and a schema-aware LLM reasoning module, filling the gap between retrieval precision and cross-document multi-entity reasoning.

- **Innovating a structured reasoning solution**: SRAG leverages structured relational tables (instead of unstructured text chunks or sparse graphs) as the core of its reasoning pipeline to address limitations of traditional RAG and graph-based RAG, with advantages of precise SQL-powered fact localization, native multi-hop entity joins, and reduced LLM attention diffusion.

- **Validating effectiveness via rigorous experiments**: SRAG significantly outperforms state-of-the-art long-context LLMs and RAG systems, achieving 27.2% and 27% accuracy improvements on MEBench and Loong respectively. These results empirically confirm the value of structured data representation in enhancing complex reasoning and answer precision for MDMEQA.

## 2 RELATED WORK

### 2.1 RETRIEVAL MECHANISMS WITH LLMS

The integration of retrieval mechanisms with large language models has been a cornerstone in advancing open-domain question answering (QA). Early RAG frameworks, pioneered by (Lewis et al., 2020), demonstrated the value of combining dense passage retrieval with generative models, but their efficacy diminishes in multi-entity scenarios where answers require synthesizing fragmented information across diverse documents. Subsequent refinements, such as REALM (Arora et al., 2023) and FiD (Izacard & Grave, 2021), improved retrieval precision through cross-attention mechanisms, yet they inherently treat documents as isolated units, failing to model inter-entity relationships critical for questions like "Compare the research contributions of Turing Award winners in the last decade." While recent long-context LLMs (*e.g.,* Claude 3 (Anthropic, 2024), GPT-4 Turbo (Achiam et al., 2023)) expand input windows to process hundreds of pages, empirical studies (Liu et al., 2025) reveal their tendency to "overlook" critical details in lengthy texts—a phenomenon termed contextual dilution—where key entities are lost due to attention saturation. Hybrid approaches, such as iterative retrieval with self-correction (Yoran et al., 2024) and hierarchical summarization chains (Wang et al., 2023), partially mitigate these issues but remain constrained by their linear processing of unstructured text, which obscures latent relational patterns between entities.

### 2.2 STRUCTURING AUGMENTED GENERATION WITH LLMS

Structured representation learning has emerged as a parallel strategy to enhance LLM reasoning. Methods like TableLLM (Zhang et al., 2025) pre-train models on tabular data to improve schema comprehension, while GraphRAG (Edge et al., 2024) constructs knowledge graphs from retrieved snippets to enable relation-aware reasoning. However, these approaches either depend on pre-defined schemas—limiting adaptability to novel domains—or suffer from computational overhead when dynamically extracting entities from heterogeneous sources, which is similar in the case of StructRAG (Li et al., 2024). Crucially, they treat structure creation as a post-retrieval step, decoupled from the initial information gathering process. In contrast, knowledge graph embedding techniques (*e.g.,* TransE (Bordes et al., 2013)) and template-based table generation prioritize static knowledge bases, rendering them ineffective for open-domain QA over evolving corpora like Wikipedia. The proposed SRAG framework uniquely addresses these gaps by unifying retrieval and structuring: it dynamically organizes extracted entities into relational tables during the retrieval phase, eliminating schema dependency through adaptive column induction (*e.g.,* inferring "field of study" and "publication count" columns for academic entity queries). This paradigm shift aligns with cognitive theories of "structure-first" reasoning (Nyamsuren & Taatgen, 2014), where tabular representations reduce LLMs' inferential burden by externalizing relational logic, thereby enabling precise aggregation of cross-document insights.

## 3 STRUCTURED RETRIEVAL-AUGMENTED GENERATION

The SRAG framework is designed to overcome the inherent challenges of multi-document, multi-entity question answering by shifting from unstructured text retrieval to structured, schema-based reasoning. As depicted in Figure 1, the system architecture consists of two core, cascaded modules: the **SQL-driven Extraction-Retrieval** module and the **Schema-aware LLM Reasoning** module. These two modules are the core components enabling Structured Retrieval-Augmented Generation, responsible for transforming unstructured documents into structured data and performing precise reasoning based on structured data, respectively. This forms a two-stage pipeline that follows the principle of "prepare the data first, then perform intelligent reasoning".

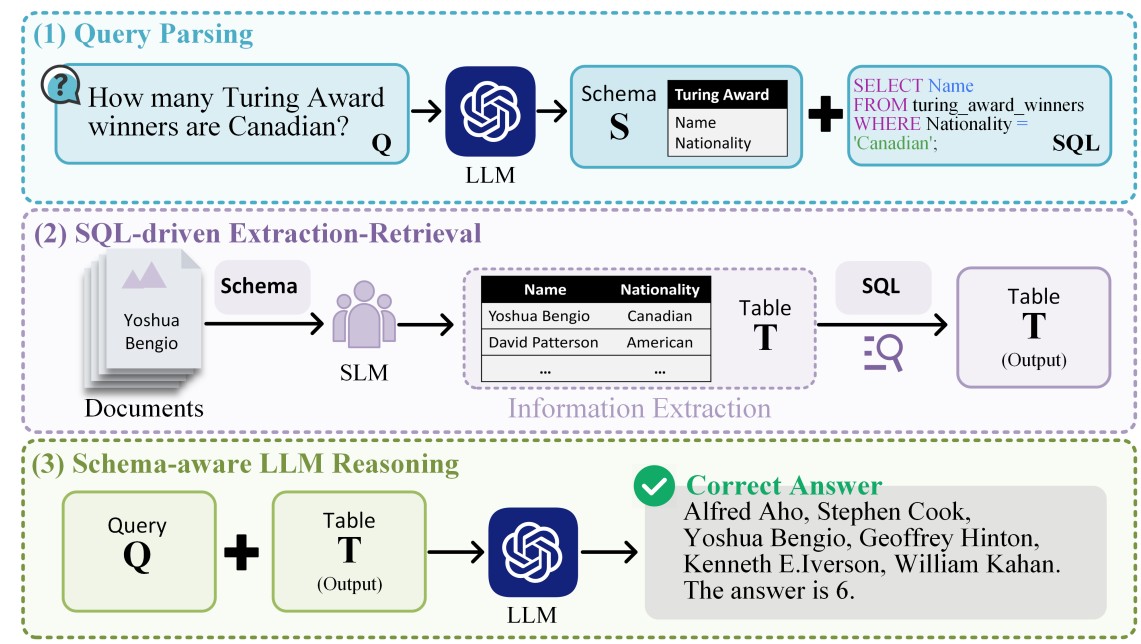

Figure 1: The overview of SRAG framework, including an SQL-driven Extraction-Retrieval module to convert unstructured text from multiple documents into tables and the Schema-aware LLM Reasoning module to infer the final answer.

## 3.1 SQL-DRIVEN EXTRACTION-RETRIEVAL MODULE

This module serves as the "structured transformation engine" of SRAG. Its core function is to convert unstructured text scattered across multiple documents into a semantically coherent relational table. Guided by the structured logic of SQL as a "blueprint", it accurately locates entities, attributes, and corresponding values relevant to the query, resolving the issue of critical fact omission inherent in traditional RAG that relies on vector similarity-based coarse-grained retrieval. The workflow consists of four steps:

**Parse the Query, Generate "Instructions" (Query → SQL & Schema).** When a user inputs a natural language question (e.g., "What were the prices of the phones released by Apple and Samsung in September 2023?"), this module first uses a LLM as a parser to analyze this question. More details are provided in the Appendix A.1.1. It generates:

- An SQL query statement: This is like an instruction given to the database. For example: SELECT company, product, price FROM release_events WHERE date = "2023-09" AND company IN ("Apple", "Samsung").
- A target table schema: This defines what kind of table we need to store the related information, including column names (*e.g.,* company, product, price) and data types.

**Multi-Task Information Extraction Based on the Schema.** To populate the target schema, we designed a multi-task information extraction framework powered by a small-scale Language Model (SLM). Rather than processing text in an unstructured manner, the SLM performs several parallel extraction tasks, such as entity recognition, attribute extraction, and relation linking, all guided by the schema. The SLM's task is very clear:

like filling in blanks, it only extracts information that matches the schema defined in the previous step and ignores all other irrelevant text. This is much more efficient and accurate than having the model understand the entire text.

**Execute SQL, Filter, and Build the Table.** All the extracted raw information (potentially many entries) is temporarily stored in a structured format. Then, execute the SQL query generated in the first step. This SQL statement automatically filters out information that doesn't meet the conditions (*e.g.,* records not from September 2023, or not from Apple or Samsung), retaining only the most precise and relevant data rows. Finally, the output is a clean, tidy, and highly relevant structured table.

**Output.** The final product of this module is not chunks of text, but an relational table. This table contains all the core facts needed to answer the user's question, and preliminary logical connections have been established through the table's structure.

The SQL-Driven Extraction-Retrieval module offers distinct advantages over traditional vector-based retrieval methods. Its precision is significantly enhanced by utilizing SQL and schema for information extraction, which is far more accurate than relying on vector similarity and effectively avoids the omission of key information. Furthermore, the process is highly efficient as the small-scale Language Model is tasked only with extracting specific information, resulting in lower computational costs and higher speed. Additionally, the module natively supports complex multi-hop queries, as SQL's JOIN operations can seamlessly connect entity relationships scattered across different documents, making it particularly powerful for multi-document reasoning.

## 3.2 SCHEMA-AWARE LLM REASONING MODULE

The second module doesn't let the LLM find answers and logic from lengthy text like "looking for a needle in a haystack". Instead, make it act like a smart data analyst, performing deterministic reasoning directly based on the structured table (schema) produced by the previous module. The module consists of two steps:

**Structured Context Injection.** The system carefully crafts a prompt that provides the Large Language Model with both the user's original question and the answer table generated by the previous module. The prompt explicitly instructs the LLM: "Please answer the question based solely on the data in the following table." This forces the LLM into a "schema-aware" state, focusing its attention entirely on the table rather than relying on potentially unreliable internal knowledge. More details are provided in the Appendix A.1.3.

**Structured Reasoning and Answer Generation.** Leveraging the structured table, the LLM's reasoning process becomes highly reliable and straightforward. It can directly sort events chronologically using the "Date" column, and correlate information across rows and columns to determine relationships—such as which product belongs to which company or which event corresponds to a specific time. By following a clear chain of thought, the LLM derives accurate answers.

The Schema-Aware LLM Reasoning Module offers significant advantages by eliminating attention diffusion, as the LLM no longer needs to process noisy and redundant information from lengthy texts, drastically improving its focus. Furthermore, it compensates for inherent LLM weaknesses by transforming tasks requiring precise calculation and logical operations, areas where LLMs typically struggle, into strengths, leveraging their capabilities in pattern recognition and instruction following. Consequently, this approach ensures that answers are reliable and traceable, since the reasoning process is grounded in clear, tabular data, making conclusions more trustworthy and easily verifiable.

These two modules are tightly integrated. The first module is responsible for accurately acquiring facts by converting unstructured text into structured data, while the second module performs efficient reasoning to generate reliable answers based on the structured data. Ultimately, this significantly enhances both the reasoning efficiency and answer accuracy in multi-document, multi-entity question answering.

# 4 EXPERIMENT

## 4.1 EXPERIMENT SETUP

**Evaluation Datasets.** We evaluated our proposed method on two challenging multi-document question-answering benchmarks: MEBench (Lin et al., 2025) and Loong (Wang et al., 2024). MEBench is a specialized benchmark for multi-entity QA, comprising 4,780 methodically crafted questions. These are systematically categorized into three primary types: Comparison, Statistics, and Relationship which aims to provide comprehensive coverage of diverse and realistic multi-entity reasoning scenarios. Loong includes four distinct reasoning tasks: Spotlight Locating, Comparison, Clustering, and Chain of Reasoning, across four increasing document length settings. This design specifically tests a model's ability to locate and connect relevant information as it becomes more dispersed throughout longer documents.

**Implementation Details.** In SQL-driven Extraction-Retrieval module, we parsed the question using GPT-4o (Achiam et al., 2023). For small-scale language model of information extraction, we used Mistral-7B (Jiang et al., 2023). In the Schema-aware LLM Reasoning module, we used GPT-4o as the reasoning model.

**Baselines.** We selected baseline methods from widely adopted and state-of-the-art approaches in MDMEQA. Among proprietary large language models , we selected the widely recognized GPT-4o (Achiam et al., 2023), which serves as a strong standalone generative baseline. To evaluate retrieval-augmented strategies, we incorporated the standard RAG framework (Lewis et al., 2020), which segments documents into short chunks and uses a retriever to select the most relevant segments based on the input question. These are then used as context for GPT-4 during answer generation. We also included two advanced structured retrieval methods: (1) GraphRAG (Edge et al., 2024), which constructs a knowledge graph from extracted (head, relation, tail) triples and uses graph retrieval and reasoning to enhance answer generation. (2) StructRAG (Li et al., 2024), a structure-aware framework that dynamically identifies suitable structured representations for a given task, reconstructs textual content into that format, and performs inference over the organized data. This selection enables a comprehensive comparison across plain LLMs, naive retrieval, and more sophisticated graph-based or structure-aware augmentation techniques.

**Evaluation Metrics.** For the MEBench benchmark, we employ Accuracy as the primary evaluation metric to measure performance on the tasks. Within the Statistics category—specifically for the sub-tasks of Variance Analysis, Correlation Analysis, and Distribution Compliance, we evaluate the correctness of the selected columns and methods. For the Loong benchmark, we adhere to the original evaluation protocol and utilize the official evaluation code. Performance is measured using a dual mechanism: LLMs are prompted to output a confidence score between 0 and 100, and final answers are also evaluated via Exact Match (EM) rate to ensure precise alignment with ground-truth responses.

## 4.2 MAIN RESULTS

Table 1 presents experimental results alongside overall accuracy on MEBench, and Table 2 shows LLM-judged scores and exact match rate in Loong benchmark. The left indicator represents the Avg Scores (0-100), and the right one represents the Perfect Rate (0-1). The experimental results demonstrate that the proposed SRAG method achieves state-of-the-art performance across both the MEBench and Loong benchmarks, significantly outperforming all baseline models. The superiority of SRAG is consistent across all question types, multi-entity reasoning tasks, and document length settings, highlighting its robustness and effectiveness in handling complex MDMEQA scenarios.

**MEBench Results.** MEBench tests a model's ability to reason over multiple entities through Comparison, Statistics, and Relationship questions.

Table 1: Experimental results for MEBench.

| Method | Accuracy | | | |
|---|---|---|---|---|
| | Comparison | Statistics | Relationship | Overall |
| **All sets** | | | | |
| GPT-4o | 0.262 | 0.353 | 0.407 | 0.338 |
| GPT-4o + RAG | 0.696 | 0.579 | 0.593 | 0.620 |
| GraphRAG | 0.618 | 0.558 | 0.593 | 0.586 |
| StructRAG | 0.678 | 0.588 | 0.573 | 0.612 |
| **SRAG (Ours)** | **0.934** | **0.908** | **0.812** | **0.892** |
| **Set1 (0-10)** | | | | |
| GPT-4o | 0.467 | 0.595 | 0.571 | 0.548 |
| GPT-4o + RAG | 0.870 | 0.690 | 0.755 | 0.764 |
| GraphRAG | 0.774 | 0.761 | 0.694 | 0.748 |
| StructRAG | 0.838 | 0.773 | 0.735 | 0.784 |
| **SRAG (Ours)** | **0.968** | **0.929** | **0.837** | **0.918** |
| **Set2 (11-100)** | | | | |
| GPT-4o | 0.388 | 0.505 | 0.525 | 0.473 |
| GPT-4o + RAG | 0.777 | 0.613 | 0.667 | 0.679 |
| GraphRAG | 0.714 | 0.589 | 0.707 | 0.659 |
| StructRAG | 0.793 | 0.601 | 0.657 | 0.676 |
| **SRAG (Ours)** | **0.952** | **0.923** | **0.818** | **0.906** |
| **Set3 (>100)** | | | | |
| GPT-4o | 0.153 | 0.214 | 0.306 | 0.219 |
| GPT-4o + RAG | 0.508 | 0.350 | 0.413 | 0.415 |
| GraphRAG | 0.450 | 0.344 | 0.417 | 0.396 |
| StructRAG | 0.492 | 0.374 | 0.359 | 0.406 |
| **SRAG (Ours)** | **0.946** | **0.884** | **0.791** | **0.879** |

- Performance of SRAG: SRAG achieved a remarkable overall accuracy of 89.2%, which is a substantial improvement over the next best method, GPT-4o + RAG (62.0%), by 27.2 percentage points. This indicates a fundamental advancement in multi-entity reasoning capabilities.

- Consistency Across Question Types: The superiority of SRAG is consistent across all question categories, with accuracies of 93.4% (Comparison), 90.8% (Statistics), and 81.2% (Relationship). This suggests that the method's underlying architecture is well-suited for the distinct logical demands of each question type.

- Robustness to Increasing Entity Density: A key finding is SRAG's exceptional robustness as the number of entities increases. While all methods see a performance drop from Set1 (0-10 entities) to Set3 (>100 entities), the decline for SRAG is minimal (from 91.8% to 87.9%). In contrast, competitors like GPT-4o + RAG and StructRAG suffer severe degradation (e.g., GPT-4o + RAG drops from 76.4% to 41.5%). This demonstrates SRAG's superior ability to locate and synthesize entity information from a large, dispersed set of documents, which is a critical requirement for real-world multi-document QA.

**Loong Results.** Loong evaluates a model's capability for specific reasoning tasks under the challenge of increasing document length.

Table 2: Experimental results for Loong.

| Method | Spotlight Locating | | Comparison | | Clustering | | Chain of Reason | | Overall | |
|---|---|---|---|---|---|---|---|---|---|---|
| **All sets** | | | | | | | | | | |
| GPT-4o | 76.79 | 0.65 | 50.98 | 0.29 | 45.04 | 0.10 | 57.46 | 0.27 | 54.17 | 0.26 |
| GPT-4o + RAG | 64.04 | 0.44 | 41.85 | 0.26 | 35.37 | 0.03 | 41.62 | 0.19 | 43.05 | 0.18 |
| GraphRAG | 22.49 | 0.00 | 22.91 | 0.01 | 37.52 | 0.03 | 45.34 | 0.23 | 33.44 | 0.07 |
| StructRAG | 68.07 | 0.40 | 63.36 | 0.36 | 60.71 | 0.14 | 53.27 | 0.18 | 60.56 | 0.23 |
| **SRAG (Ours)** | **85.06** | **0.84** | **73.28** | **0.43** | **64.43** | **0.42** | **73.67** | **0.57** | **68.29** | **0.53** |
| **10K-50K Tokens** | | | | | | | | | | |
| GPT-4o | 87.38 | 0.83 | 65.56 | 0.34 | 58.15 | 0.24 | 83.21 | 0.56 | 71.81 | 0.45 |
| GPT-4o + RAG | 50.57 | 0.35 | 44.08 | 0.27 | 37.58 | 0.05 | 53.41 | 0.35 | 45.65 | 0.23 |
| GraphRAG | 32.30 | 0.02 | 28.15 | 0.03 | 41.52 | 0.14 | 55.38 | 0.44 | 41.64 | 0.18 |
| StructRAG | 76.02 | 0.48 | 77.09 | 0.48 | 66.43 | 0.23 | 69.20 | 0.35 | 70.82 | 0.36 |
| **SRAG (Ours)** | 91.12 | 0.94 | 87.10 | 0.58 | 67.97 | 0.45 | 90.84 | 0.70 | 80.09 | 0.62 |
| **50K-100K Tokens** | | | | | | | | | | |
| GPT-4o | 88.50 | 0.73 | 61.01 | 0.41 | 48.79 | 0.11 | 63.33 | 0.35 | 59.55 | 0.30 |
| GPT-4o + RAG | 67.60 | 0.47 | 47.21 | 0.32 | 39.73 | 0.05 | 47.07 | 0.22 | 46.33 | 0.19 |
| GraphRAG | 24.55 | 0.00 | 14.15 | 0.00 | 37.48 | 0.00 | 45.79 | 0.12 | 32.73 | 0.03 |
| StructRAG | 69.36 | 0.42 | 64.98 | 0.37 | 62.63 | 0.17 | 55.79 | 0.19 | 62.17 | 0.24 |
| **SRAG (Ours)** | 88.79 | 0.88 | 75.68 | 0.50 | 63.19 | 0.42 | 71.23 | 0.55 | 70.76 | 0.54 |
| **100K-200K Tokens** | | | | | | | | | | |
| GPT-4o | 76.34 | 0.66 | 43.25 | 0.21 | 39.47 | 0.04 | 45.96 | 0.09 | 47.89 | 0.19 |
| GPT-4o + RAG | 75.16 | 0.56 | 43.04 | 0.28 | 33.44 | 0.02 | 39.53 | 0.14 | 44.73 | 0.18 |
| GraphRAG | 16.15 | 0.00 | 27.95 | 0.00 | 43.35 | 0.00 | 44.20 | 0.17 | 33.95 | 0.04 |
| StructRAG | 67.25 | 0.43 | 56.59 | 0.34 | 57.10 | 0.10 | 48.74 | 0.13 | 56.76 | 0.21 |
| **SRAG (Ours)** | 81.44 | 0.80 | 68.12 | 0.30 | 61.72 | 0.41 | 68.15 | 0.52 | 62.43 | 0.48 |
| **200K-250K Tokens** | | | | | | | | | | |
| GPT-4o | 37.53 | 0.19 | 24.45 | 0.08 | 31.01 | 0.00 | 33.55 | 0.07 | 31.73 | 0.07 |
| GPT-4o + RAG | 53.21 | 0.24 | 24.85 | 0.10 | 27.05 | 0.00 | 17.97 | 0.00 | 29.58 | 0.07 |
| GraphRAG | 17.85 | 0.00 | 27.20 | 0.00 | 21.33 | 0.01 | 34.15 | 0.34 | 23.94 | 0.05 |
| StructRAG | 58.01 | 0.19 | 56.73 | 0.26 | 57.72 | 0.00 | 36.42 | 0.05 | 52.45 | 0.10 |
| **SRAG (Ours)** | 72.86 | 0.71 | 61.65 | 0.30 | 70.34 | 0.39 | 69.85 | 0.54 | 60.52 | 0.47 |

- Superior Overall Performance: SRAG achieves the highest Overall Avg Score (68.29) and, more importantly, a dramatically higher Perfect Rate (0.53) compared to all other methods. The Perfect Rate metric is a stringent indicator of how often a model produces a fully correct answer. SRAG's 0.53 rate is more than double that of the next best model (GPT-4o at 0.26), highlighting its precision and reliability.

- Task-specific Strengths: SRAG excels in tasks requiring precise information location and complex reasoning. It leads in Spotlight Locating (0.84) and Chain of Reasoning (0.57), demonstrating an unmatched ability to find key facts and perform multi-step inferences. It also performs strongly in Comparison and Clustering.

- Handling Long Document Contexts: The results across increasing token lengths confirm SRAG's scalability. While all models struggle with the longest documents (200K-250K tokens), SRAG's performance decline is the least severe. It maintains a strong Avg Score of 60.52 and a dominant Perfect Rate of 0.47 in the most challenging setting, whereas other models see their Perfect Rates

drop to 0.10 or below. This proves that SRAG's method for structuring information is crucial for managing the complexity and information dispersion inherent in long-context scenarios.

### 4.3 ANALYSIS OF RESULTS

The design of our system inherently incorporates the logic of an ablation study. The system comprises two core modules. For the first module, the SQL-Driven Extraction-Retrieval Module, alternative components such as a vanilla LLM, RAG, and GraphRAG can be directly substituted and compared. Similarly, for the second module, the Schema-Aware LLM Reasoning Module, alternative components like a vanilla LLM and StructRAG serve as comparable replacements. Therefore, the comparative experiments presented in the main results (e.g., comparing SRAG against GPT-4o, GPT-4o+RAG, GraphRAG, and StructRAG) effectively function as a comprehensive ablation study. By evaluating these different combinations, the performance contribution of each proposed module is directly assessed against its ablated counterparts, making a dedicated ablation experiment unnecessary.

The experimental evaluation on two challenging multi-document QA benchmarks leads to the following conclusions:

- Significant Advancement: The proposed SRAG method establishes a new state-of-the-art for multi-document question answering, significantly outperforming strong baselines.
- Robustness: The most notable advantage of SRAG is its robustness to scale. Its performance remains consistently high even as the number of entities or the length of the context increases dramatically, a scenario where other models fail significantly.
- Effective Reasoning: SRAG demonstrates superior capabilities across a diverse set of reasoning tasks, from simple fact location (Spotlight) to complex, multi-step reasoning chains. Its high performance on Loong indicates that it produces correct answers more reliably and completely.

These results strongly validate the design principles of SRAG, suggesting that its structured approach to organizing and reasoning over information from multiple documents is highly effective for tackling the challenges of real-world, large-scale question answering.

## 5 CONCLUSION

This paper addresses the core challenges of Multi-document Multi-entity Question Answering (MDMEQA), where LLMs and traditional RAG frameworks struggle to build cross-document evidence chains, deduce entity relationships from fragmented information, omit critical facts due to coarse-grained retrieval, and inefficiently integrate complex relationship networks in multi-document scenarios, by proposing Structured Retrieval-Augmented Generation (SRAG), a two-stage framework that first converts unstructured text into semantically coherent relational tables via a SQL-driven Extraction-Retrieval module and then guides LLMs toward schema-aware relational reasoning. SRAG offers three key advantages: precise fact localization enabled by SQL-powered indexing, native support for multi-hop entity join operations through relational tables, and mitigation of LLMs' attention diffusion effect via structuring. Empirical evaluations on MDMEQA benchmarks MEBench and Loong show SRAG significantly outperforms state-of-the-art long-context LLMs and RAG systems, highlighting the pivotal role of structured data representation in enhancing complex reasoning and answer precision for MDMEQA; the work also lays a foundation for future advancements, such as extending SRAG to diverse document types, optimizing it for low-resource domains, and exploring multi-modal relational reasoning.

**Limitations.** While the SRAG framework demonstrates superior performance, several avenues for future work remain. This work focuses on establishing the conceptual advantage of structured retrieval. Some practical challenges, such as further reducing the retrieval latency for very large corpora and adapting the table schema to highly domain-specific terms, are noted as interesting points for future exploration.

**Ethics Statement.** All experiments in this work are conducted on public benchmarks, which contain no private or sensitive information. The proposed SRAG framework provides a general methodology for enhancing reasoning in multi-document QA by structuring textual data. We advocate for the responsible use of this technology, emphasizing the necessity of domain-specific validation to ensure factual correctness and mitigate biases that may exist in the underlying language models or source documents.

**Reproducibility Statement.** We have made significant efforts to ensure the reproducibility of our work. The full source code, along with the data processing and evaluation scripts, has been made publicly available at https://anonymous.4open.science/r/SRAG-07A7. The repository includes detailed instructions for replicating the two-stage SRAG pipeline. All hyperparameters and implementation details necessary to reproduce the experimental results are provided in the codebase and the supplementary materials.

**Statement on LLM Usage.** Large Language Models (LLMs) are used in this research strictly for auxiliary and supportive tasks. These tasks include polishing the writing for improved clarity and fluency, and assisting with minor code debugging during implementation. The core research design, novel methodology (SRAG framework), algorithmic development, and substantive writing of the paper are the original work of the authors. Accordingly, the use of LLMs does not constitute a substantive intellectual contribution to the key findings or contributions of this work.

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

# A APPENDIX

## A.1 PROMPT

### A.1.1 PROMPT FOR SCHEMA

Instruction:
You are an expert data analyst and SQL architect. Your task is to analyze the natural language question and generate two precise outputs:
Input Question: {question}
Your Task:
1. Generate an SQL Query Statement: Create a syntactically correct SQL query that would retrieve the answer from a relational database. This SQL should function as precise instructions to the database system.
2. Define Target Table Schema:
Design the structure of the table needed to store the answer, including: Column names (e.g., company, product, price); Appropriate data types for each column; Any relevant constraints or key specifications. Please provide your response in the following exact format:
SQL Query:
[Your SQL query here]
Target Schema:
- column1: data_type [constraints]
- column2: data_type [constraints]
- column3: data_type [constraints]
Guidelines: Base the SQL query on common database patterns that would logically contain the required information. Ensure the target schema accurately represents the structure of the query results. Use appropriate SQL standards and data types (VARCHAR, INT, DECIMAL, DATE, etc.). Consider the relationships and entities mentioned in the question.

### A.1.2 PROMPT FOR SLM

Instruction:
You are a precise information extraction agent. Your task is to extract specific pieces of information from the provided text according to the target schema. Treat this as a structured filling task - only extract what matches the schema definition.
Target Schema:
{Schema}
Text:
{Source Text}
Guidelines: Extract ONLY information that directly corresponds to the schema elements; Ignore all text that doesn't match the schema requirements; If information for a schema element is not found, return null/empty; Treat this as a blank-filling exercise rather than text comprehension; Be precise and literal in your extractions.
Output Format:
Return a structured JSON object that mirrors the schema.
Critical Constraint:
Do not interpret, infer, or add any information not explicitly present in the text. Your role is purely extractive, not generative.

### A.1.3 PROMPT FOR STRUCTURED CONTEXT INJECTION

> Please answer the question based solely on the data in the following table. Do not rely on any internal or external knowledge outside of the table provided. Your response must be derived exclusively from the table data.
> Question:
> {The question }
> Table:
> {The table generated by the previous module }
> Remember: Your answer should be grounded only in the information presented in the table above.

## A.2 OPTIMIZATION

Two aspects of optimization are included in SRAG system to enhance the overall performance:

**Model Selection.** Model selection is straightforward yet highly effective for optimization (Liu et al., 2024). The SRAG system comprises multiple tasks, necessitating the selection of the most suitable model for different tasks. For basic tasks, more affordable and faster LLMs can suffice, while utilization of the most advanced LLMs is essential in more complex tasks to ensure optimal performance. Specifically, SRAG system employs powerful yet resource-intensive GPT-4o for tasks such as generation of table schema and SQL queries. In contrast, for more basic information extraction, we utilize open-source Mistral-7B, thereby achieving a balance between cost efficiency and functional performance.

**LLM Input/Output Control**. SplitWise (Patel et al., 2023) shows that LLM inference time is generally proportional to the size of input and output tokens. Since GPT models decide the cost based on the input token, we try to minimize the input of large models. Meanwhile, we use the instructive prompt to reduce the size of the outputs generated by LLM without changing the quality of these outputs. The example of prompt is in Appendix A.2.1.

### A.2.1 PROMPT FOR OUTPUT CONTROL

> ...
> review your output to ensure it meets all the above criteria. Your goal is to produce a clear, accurate, and well-structured output. Just output the {}, no other word or symbol.

