# OpenReview forum: "Structured Retrieval-Augmented Generation for Multi-Doc Multi-Entity Question Answering"
_ICLR.cc/2026/Conference — ICLR 2026 Conference Withdrawn Submission_

### Official Review · Reviewer_d8Ti · 2025-10-29

**Soundness:** 1
**Presentation:** 2
**Contribution:** 1
**Rating:** 2
**Confidence:** 4

**Summary:**

This paper addresses the challenges of Multi-document Multi-entity Question Answering (MDMEQA), where existing RAG frameworks struggle to construct cross-document evidence chains, locate critical facts, and reason over fragmented entity relationships.
The authors propose Structured Retrieval-Augmented Generation (SRAG), a two-stage framework:
(1) a SQL-driven Extraction-Retrieval module that converts unstructured text from multiple documents into relational tables, and
(2) a schema-aware LLM reasoning module that uses these tables to perform targeted, structured reasoning.

The paper evaluates SRAG on two MDMEQA benchmarks (MEBench and Loong), reporting 27.2% and 27% accuracy improvements over state-of-the-art (SOTA) long-context LLMs and RAG systems, respectively.
The authors highlight three key advantages of SRAG: precise fact localization via SQL indexing, native support for multi-hop entity joins, and mitigation of LLM attention diffusion.

**Strengths:**

1. This paper proposes a neuro-symbolic reasoning approach to address multi-entity, multi-document question answering tasks in long texts. The authors formalize queries by parsing questions into executable SQL statements, utilizing SQL as an intermediate representation and leveraging large language models for structure-aware reasoning.

2. The authors conduct experiments on two long-context with multi-entity and multi-document datasets and validates the effectiveness of the proposed methods.

**Weaknesses:**

1. The method proposed in this paper is essentially a neuro-symbolic reasoning approach that utilizes SQL as an intermediate symbolic language, combining the neural reasoning of large language models with the symbolic reasoning of SQL query language. This type of neuro-symbolic reasoning method has already been widely adopted in structured reasoning tasks, such as KBQA and logical reasoning. The approach presented in this paper applies this neuro-symbolic reasoning to the multi-document multi-entity scenario. Thus, the primary contribution lies in domain adaptation, while the methodology largely follows existing frameworks, limiting its novelty.
Besides, the authors did not cite any neural-symbolic methods in this manuscrip.

2. The method employs SQL as an intermediate representation, a process that inevitably introduces information loss and errors. The extent and impact of such errors critically constrain the effectiveness of the proposed method. The authors should systematically quantify the errors introduced by this intermediate transformation to clarify the reliability of their approach.

3. The task selected in this study is multi-entity multi-document question answering. To what extent does this task overlap with multi-hop question answering (MHQA)? Why did the authors exclude multi-hop QA datasets in their experiments? The evaluation relies on only two datasets with limited domain coverage, making the experimental validation relatively insufficient.

**Questions:**

Please refer to weaknesses section.

**Details Of Ethics Concerns:**

It seems that this paper doesn't use the correct ICLR style format.
It appears to have significantly smaller margins than the official template.

---

### Official Review · Reviewer_DDiY · 2025-10-31

**Soundness:** 1
**Presentation:** 2
**Contribution:** 2
**Rating:** 2
**Confidence:** 4

**Summary:**

The paper addresses the problem of multi-document multi-entity question answering (MDMEQA), where an answer has to be generated by retrieving and finding relationships between natural language document.. The authors claim that traditional RAG approaches struggle to retrieve correct information because they rely solely on vector similarity. They also claim that graph-based RAGs struggle with efficient retrieval, as they require traversing long paths within graph structures.
    To overcome these challenges, the paper proposes the Structured Retrieval-Augmented Generation (SRAG) framework, consisting of two modules. (1) The SQL-driven Extraction–Retrieval Module constructs query-specific tables from the input documents. It first generates an SQL query and target table schemas based on the input question. Then, it extracts records from the candidate documents that match those schemas using a small LLM. By executing the SQL query on these extracted tables, a query-relevant table is finally produced. (2) The Schema-aware LLM Reasoning Module simply prompts an LLM with the resulting table and the user’s question to generate a final answer.
    The authors evaluate SRAG on two benchmarks, MEBench and Loong, showing that SRAG outperforms state-of-the-art accuracy on both benchmarks. However, they omit comparison with a stronger baseline for the Loong benchmark that outperforms SRAG. Although they also claim that SRAG is efficient, no experiments are  provided to substantiate this claim.

**Strengths:**

S1. The authors empirically demonstrate the structured RAG’s effectiveness in a specific experimental setting.

S2. The paper is clearly written and not hard to follow.

**Weaknesses:**

W1. Several key claims and motivations are insufficiently substantiated.

W1.1. The authors should provide empirical evidence to support their claim that the SQL-driven extraction–retrieval module is efficient. While they claim in Line 203 that “the process is highly efficient as the small-scale language model is tasked only with extracting specific information, resulting in lower computational costs and higher speed.”, it is not backed up by any empirical evidence.

W1.2. The introduction (line 55) states that multi-document settings create “scattered, overlapping relationship networks, making graph traversal inefficient” and that multi‑hop reasoning over graphs becomes “computationally prohibitive.” This is a key motivation, but remains neither theoretically justified nor empirically validated. The authors should include complexity analyses or runtime experiments.

W1.3. The paper attributes MDMEQA failures to an “attention diffusion effect,” yet provides neither direct citation nor quantitative evidence for this claim. Please include empirical validation or supporting references.

W2. The experimental evaluation is incomplete, and some results reveal important limitations of SRAG.

W2.1. A state-of-the-art competitor on Loong is missing [1]. The model reports an Overall AS of 75.56 and an Overall PR of 0.57 on Loong, outperforming SRAG’s corresponding scores of 68.29 and 0.53. It would also be necessary to include this competitor in the evaluation on MEBench for completeness.

W2.2. SRAG appears to show poor generalizability based on the reported accuracy results. While SRAG achieves a high accuracy of 0.892 on MEBench, its performance drops to 0.53 on Loong. The discrepancy may stem from the fact that  MEBench’s question-answer generation process aligns closely with SRAG’s table-extraction mechanism, as MEBench constructs its QA pairs directly from tables generated from text. In contrast, Loong lacks this alignment, which likely SRAG’s weaker performance -- further supported by the existence of a competitor that outperforms SRAG (see W2.1).

W2.3. The survey of applicable multi-hop retrieval methods is insufficient, and these should be added as competitors. For example,  [2] presents a method for multi-hop retrieval using only LLMs without training. Whereas the authors use a RAG framework from 2020 as their baseline, which is too outdated.

[1] Lin, Jingyang, et al. "Facilitating long context understanding via supervised chain-of-thought reasoning." arXiv preprint arXiv:2502.13127 (2025).

[2] Zhang, Xiaoming, et al. "Hierarchical retrieval-augmented generation model with rethink for multi-hop question answering." arXiv preprint arXiv:2408.11875 (2024).

W3. Novelty is insufficient.

W3.1. The SQL-driven Extraction-Relation Module, which serves as a major contribution, seems to be a straightforward adaptation of existing research. (1) Extracting documents into structured data has already been studied in the multimodal RAG domain. For example, [3] extracts structured data of triples from documents containing both text and image. SRAG merely modifies this approach by incorporating query information to perform on-the-fly extraction and by generating records instead of triples. (2) Retrieval using SQL is a direct application of prior text-to-SQL research [4]. Moreover, the use of the most naive LLM-driven text-to-SQL further reduces novelty.

W3.2. The Schema-aware LLM Reasoning Module adds little innovation, as it simply involves prompting the LLM with the retrieved table.

[3] Yang, Qian, et al. "Enhancing multi-modal multi-hop question answering via structured knowledge and unified retrieval-generation." Proceedings of the 31st ACM International Conference on Multimedia. 2023.

[4] Hong, Zijin, et al. "Next-generation database interfaces: A survey of llm-based text-to-sql." IEEE Transactions on Knowledge and Data Engineering (2025).

W4. The proposed method appears to have scalability issues. The SQL-driven Extraction-Retrieval Module appears computationally heavy, as the SLM has to be invoked online for each document in the data source to perform information extraction. The absence of runtime experiments or complexity analysis further intensifies this concern.

W5. The experiments lack sufficient explanation and analysis.

W5.1. A clear explanation is needed for why RAG-based models underperform compared to non-RAG models on Loong. According to the authors’ claims, the use of  RAG should reduce “attention diffusion” and improve accuracy. A deeper analysis is required for this counter-intuitive experimental result.

W5.2. In the experiment setups of Section 4.1 , more details should be added regarding the “GPT-4o + RAG”. What is the length of each chunk? Which embedding model is used (and how large is the performance gap from the state-of-the-art embedding model Qwen3-Embedding-8B [5])? How many chunks does it retrieve?

[5] Zhang, Yanzhao, et al. "Qwen3 Embedding: Advancing Text Embedding and Reranking Through Foundation Models." arXiv preprint arXiv:2506.05176 (2025).

W6. The paper’s readability and completeness need some improvement.

W6.1. The extent of heuristic usage should be reported. I have found that several fallback heuristics (Line 111, Line 149) exist in the shared code `srag/sqlextractionretrieval.py`; they should be explained within the paper, and the authors should report the ratio of questions where these heuristics are invoked.

W6.2. The authors should include an explicit problem definition or problem description section. Current writing imposes ambiguity in its setting (e.g., document modality, open-domain vs distractor setting).

W6.3. The authors should thoroughly explain or cite undefined concepts (e.g., “fact localization”).

W6.4. In the abstract, it should be %p rather than %.

**Questions:**

Please refer to the weaknesses.

---

### Official Review · Reviewer_ZZTL · 2025-10-31

**Soundness:** 2
**Presentation:** 2
**Contribution:** 1
**Rating:** 2
**Confidence:** 3

**Summary:**

The paper targets multi-document, multi-entity QA (MDMEQA), arguing that long-context LLMs and standard RAG fail to construct cross-document evidence chains and suffer from attention diffusion. It proposes a two-stage SRAG pipeline: (i) an SQL-driven extraction-retrieval step that converts unstructured text to a relational table; (ii) a schema-aware LLM reasoning step that answers solely from the table. Evaluations on MEBench and Loong show large gains over GPT-4o, vanilla RAG, GraphRAG and StructRAG.

**Strengths:**

1. The proposed method achieves significant performance gains.
2. Leveraging SQL to generate structured data as context for LLM is reasonable.

**Weaknesses:**

1. **Limited novelty.**
Core idea is converting text to tables earlier, then reason table-wise. Closest prior (StructRAG / GraphRAG) already explores structure-aware retrieval and inference over structured representations; the paper mainly moves structuring earlier to the retrieval phase. No learning/training or new modeling primitives are introduced. The method relies on prompted GPT-4o and a small extractor (Mistral-7B) to realize the pipeline.

2. **Lack of analysis on the cost and efficiency.**
Although the paper claims efficiency (small model for extraction; reduced attention by structured context), there is no measurement of latency, token/$$ cost, or throughput versus baselines. Given reliance on GPT-4o for both parsing and final reasoning, actual query-time cost may be high.

3. **Claims lack quantitative support.**
Most analytical claims in Section 4.3 (efficiency, reduced attention diffusion, robustness, determinism) are not backed by any quantitative experiments, ablation studies or case studies.

**Questions:**

Please refer to weaknesses.

---

### Note · Authors · 2026-01-06

I have read and agree with the venue's withdrawal policy on behalf of myself and my co-authors.